# Randomized PCA Forest for Unsupervised Outlier Detection

## Abstract

We propose a novel unsupervised outlier detection method based on Randomized Principal Component Analysis (PCA). Motivated by the performance of Randomized PCA (RPCA) Forest in approximate K-Nearest Neighbor (KNN) search, we develop a novel unsupervised outlier detection method that utilizes RPCA Forest for unsupervised outlier detection by deriving an outlier score from its intrinsic properties. Experimental results showcase the superiority of the proposed approach compared to the classical and state-of-the-art methods in performing the outlier detection task on several datasets while performing competitively on the rest. The extensive analysis of the proposed method reflects its robustness and its computational efficiency, highlighting it as a good choice for unsupervised outlier detection.

## 1 Introduction

An outlier, as defined by Hawkins (1980), is "an observation which deviates so much from other observations as to arouse suspicions that it was generated by a different mechanism." Similarly, Barnett & Lewis (1994) describe it as "an observation (or subset of observations) which appears to be inconsistent with the remainder of that set of data." Outlier detection is the process of identifying such outliers. It is one of the most important and fundamental tasks in data mining and machine learning with applications in intrusion detection (Jin et al., 2021), fault detection (Wang et al., 2023), fraud detection (Caroline Cynthia & Thomas George, 2021) and others (Garcez Duarte & Sakr, 2025; Meng et al., 2024).

In recent years, many methods have been proposed to carry out the outlier detection task (Anderberg et al., 2024; Delić et al., 2024; Du et al., 2024; Zhou et al., 2024; Li et al., 2024). Despite the demonstration of promising results, further studies show that these results might be limited only to specific instances of the problem (e.g., a limited selection of datasets, a specific kind of outliers, etc.) (Campos et al., 2016). These studies also argue that classical outlier detection methods such as K-Nearest Neighbor (KNN) (Ramaswamy et al., 2000) or Local Outlier Factor (LOF) (Breunig et al., 2000) might still be the best and most effective approaches for outlier detection in most scenarios. Challenges in outlier detection, such as the encompassment of every instance of normal behavior, imprecise boundaries between normal samples and outliers, presence of noise in normal data, and the unavailability of labeled data for training (Singh & Upadhyaya, 2012) are still addressed more thoroughly by classical methods like KNN and LOF that arguably remain state-of-the-art (Campos et al., 2016). Recently proposed algorithms might offer better performance for a specific domain or application, but they do not offer a general solution for outlier detection in the same way that KNN and LOF do. A comprehensive evaluation on a broad range of datasets must be conducted on novel outlier detection algorithms to investigate different aspects of their performance and reveal where their strengths and, more importantly, where their weaknesses lie (Campos et al., 2016; Sanchez Vinces et al., 2025).

In this paper, we propose a novel generalizable algorithm for outlier detection based on Principal Component Analysis (PCA). PCA is a well-known statistical technique used to reduce the dimensionality of a dataset while preserving most of its variance. It transforms the data into a new set of orthogonal variables called principal components, which are ordered by the amount of variance they capture. PCA is used widely in different fields of data mining and machine learning such as spectral data classification (Madden & Ryder, 2005), healthcare data analysis and disease prediction (Wolberg et al., 1995), time series data mining (Li, 2014), and dataset similarity analysis (Rajabinasab et al., 2025a). Inspired by the performance of Randomized PCA (RPCA) Forest (Rajabinasab et al., 2024) in approximate KNN search, we develop a novel

unsupervised outlier detection method by utilizing an RPCA forest. In the RPCA forest, randomized PCA (Halko et al., 2011) is used to project data into a lower-dimensional subspace that contains most of the informational value of the original feature space, preserving the information while reducing the dimensionality. By using randomized PCA, we achieve greater efficiency without noticeable loss of accuracy in the calculation of principal components (Rajabinasab et al., 2024). By finding a proper splitting criterion, we continue constructing the trees. The leaves of the trees contain most similar data points. The ensemble of the trees forms the forest which is employed to carry out the outlier detection task. We propose methods to derive an outlier score from RPCA forest which can be used to perform the outlier detection task accurately. Moreover, tree-based methods are highly efficient, with computational complexity typically scaling logarithmically with the number of data points, depending on the tree's depth (Rajabinasab et al., 2024).

The rest of this paper is organized as follows: Section 2 reviews the related literature, focusing on classical methods that continue to define the state-of-the-art in the field, while also incorporating an overview of novel and recent methodological developments to contextualize the current research landscape. Section 3 elaborates on the proposed methods, offering a detailed explanation of the computation of the outlier score. Section 4 presents the experimental evaluation, including a thorough description of the datasets used for the experiments, experimental results, different analyses, and the resulting findings. Finally, Section 5 concludes the paper by reflecting on the limitations of the study and proposing directions for future research.

## 2 Related Work

Outlier detection is a fundamental machine learning task grounding in statistics (Zimek & Filzmoser, 2018). Many solutions have been proposed for this task, addressing various challenges (Schubert et al., 2014), and new methods are continuously proposed also recently (Anderberg et al., 2024; Delić et al., 2024; Du et al., 2024; Zhou et al., 2024; Li et al., 2024). However, comprehensive studies (Campos et al., 2016; Goldstein & Uchida, 2016) showed that, despite of valid attempts conducted in every research to propose more robust and effective outlier detection methods, the classical methods such as KNN (Ramaswamy et al., 2000) and LOF (Breunig et al., 2000) still remain state-of-the-art. To provide a comprehensive overview of the related work, we divide them into three major groups, NN-based methods, Tree-based methods, and Approximate NN-based methods. This division is due to the fact that the proposed method stands somewhere in between the three. We also provide a brief overview of other advances in the outlier detection field.

### 2.1 NN-based Methods

LOF (Breunig et al., 2000) and KNN (Ramaswamy et al., 2000) are the most famous methods of a class of unsupervised outlier detection methods which are based on the K-nearest neighborhoods. KNN (Ramaswamy et al., 2000) uses the KNN distance directly for outlier detection. There are several variations of the KNN algorithm for outlier detection. For example, KNN-Weight (KNNW) (Angiulli & Pizzuti, 2002) uses the sum of distances to an object's KNNs in order to reduce variation in scores and make the score more robust against the change of the parameter $k$. Outlier Detection Using Indegree Number (ODIN) (Hautamaki et al., 2004) utilizes the KNN graph and defines outlierness as a low number of in-adjacent edges, relating to using small hubness (Radovanovic et al., 2015). One of the classical and most well-known outlier detection methods is LOF. LOF is based on a local reachability density, estimated from the K-nearest neighborhood of each point, and a comparison of these local models to the local models of the K-nearest neighborhood. LOF shows how a point differs from other observations in its vicinity. There are many variants of LOF, such as Simplified LOF (SLOF) (Schubert et al., 2014) which uses KNN distance in place of the local reachability density and hence, avoids one level of neighborhood computation. Dimensionality-Aware Outlier Detection (DAO) (Anderberg et al., 2024) utilizes Local Intrinsic Dimensionality (LID) (Houle, 2017) for outlier detection. In this method, a local estimation of LID within a neighborhood is calculated and then used alongside with the KNN distance to act as an outlierness criterion.

## 2.2 Tree-based Methods

The most fundamental and important tree-based method for outlier detection is Isolation Forest (iForest) (Liu et al., 2008). iForest randomly selects one feature for each tree and tries isolating data points (keeping only one data point in a leaf) based on the values of that feature. The method is built on the idea that outliers are isolated faster (in a lower depth) compared to the inliers. Other variants of iForest include the Extended Isolation Forest (EIF) (Hariri et al., 2019), which suggests a solution for issues with inconsistent assignment of anomaly score to given data points by iForest in special cases, and the Deep Isolation Forest (Xu et al., 2023) which introduces a new representation scheme by utilizing casually initialized neural networks and mapping original data into random representation ensembles, where random axis-parallel cuts are subsequently applied to perform the data partition. Despite the fact that iForest and its extensions follow the common procedure of tree-based methods in constructing and branching the trees, it is noteworthy that it is not the only way that a tree structure can be utilized for outlier detection. There are many other attempts, leveraging the tree-structure to conduct the outlier detection task (Papadimitriou et al., 2003; Cortes, 2020; Guha et al., 2016). The proposed method of this paper also follows a novel approach in utilizing the tree structure for deriving an outlierness criterion.

## 2.3 Approximate NN-based Methods

Many methods in outlier detection are, to some extent, based on K-nearest neighbors search, and finding the K-nearest neighbors for every point is the computational bottleneck for all these methods (Kirner et al., 2017). Therefore some methods are based on approximate neighborhood search to improve efficiency while staying close to the quality achieved with exact neighborhoods. Examples include the use of locality sensitive hashing (LSH) (Wang et al., 2011), combinations of LSH and isolation forests (Zhang et al., 2017), projection-indexed nearest neighbors (de Vries et al., 2012), other variants of random projections (Bhattacharya et al., 2021), space-filling curves (Schubert et al., 2015; Kirner et al., 2017), or Hierarchical Navigable Small World (HNSW) (Malkov & Yashunin, 2020) graphs (Okkels et al., 2024). In combination with ensemble techniques, it has also been shown that using approximate neighborhoods can actually improve the quality of the results (Kirner et al., 2017). As discussed by Okkels et al. (2024), we can distinguish a "blackbox" and a "whitebox" use of approximate neighborhood search methods in these approaches. Many methods use the approximate neighborhood search as a module that is replacing the exact search, not considering the internal structure of the search method ("blackbox"). Some others make use of internal structures, such as the graph underlying some HNSW instance (Okkels et al., 2024), thus treating the search method as "whitebox". One might argue that our method follows this "whitebox" strategy, based on RPCA forests (Rajabinasab et al., 2024).

## 2.4 Other Approaches

There are several other approaches in outlier detection, mostly semi-supervised and based on deep-learning, which despite not being closely related to the proposed method, it is still important to provide a brief review of them to highlight the advancements in the field. Delić et al. (2024) utilize a K+1-way classifier for outlier detection by formulating a novel outlier score as an ensemble of in-distribution uncertainty and the posterior of the negative class. Du et al. (2024) use a Generative Adversarial Network (GAN) for outlier detection alongside with an autoencoder. The main idea is that GANs tend to favor the majority (normal) class in order to minimize the error. The generated normal samples are used for data augmentation and training the autoencoder. The reconstruction error of the autoencoder determines the outlierness score for data samples which are fed to it in a semi-supervised fashion. Hybrid Deep Support Vector Data Description (HDSVDD) (Rajabinasab et al., 2025b) is another semi-supervised approach based on deep learning which aims to learn a low-dimensional subspace to enhance the outlier detection capabilities. Mukherjee et al. (2024), identify outliers by measuring how consistently a data point's distances are preserved when projecting data from a high-dimensional space into a lower-dimensional PCA subspace. Inliers which align with the dataset's primary manifold exhibit high variance because their distances are preserved consistently with local neighbors but compressed significantly relative to the rest of the data. In contrast, outliers do not follow the global structure, leading to erratic compression ratios across the dataset.

## 2.5 Summary

Despite many methods being proposed every year to carry out the outlier detection task, classical methods such as LOF (Breunig et al., 2000), SLOF (Schubert et al., 2014), KNN (Ramaswamy et al., 2000) and iForest (Liu et al., 2008) remain state-of-the-art (Campos et al., 2016; Goldstein & Uchida, 2016; Sanchez Vinces et al., 2025). In particular, LOF and KNN have also been found to outperform deep learning-based methods (Han et al., 2022). As a recent improvement, DAO (Anderberg et al., 2024) is a comprehensive method for outlier detection especially for datasets with high dimensionality and varying local intrinsic dimensionality and shows good versatility across various datasets.

The method proposed in this paper can be located somewhere in between the three main categories we introduced. RPCA forest is an unsupervised outlier detection method, which utilizes a tree-based method, originally meant for approximate KNN search. It follows a combination of the methodologies of all the three categories, calculating (a limited number of) distances similar to KNN-based methods, using tree structure and node depth similar to tree-based methods, and following the whitebox strategy of approximate nearest neighbor search approaches.

# 3 Randomized PCA

PCA is a well-known and established method in computer science. PCA aims to linearly transform data into a new orthonormal basis where the principal directions capture variation in the data. Each principal component is a linear combination of the original variables. PCA can be useful for dimensionality reduction and data analysis by transforming data into a representation where the elements are mutually uncorrelated. The first principal component includes the most amount of variance (information) and this amount reduces as we move towards the next components.

Let $\mathbf{X}$ be an $(n \times d)$ matrix of $n$ different observations of $d$ variables. Let $\tilde{\mathbf{X}}$ be the mean-subtracted (i.e., centered) translation of $\mathbf{X}$. The sample covariance matrix $\mathbf{S}$ can be expressed as:

$$\mathbf{S} = \frac{1}{n-1}\tilde{\mathbf{X}}^T\tilde{\mathbf{X}} \tag{1}$$

The first principal component is calculated as a linear combination of the original variables $z_{i1} = \mathbf{a}_1^T\tilde{\mathbf{x}}_i$ where $\tilde{\mathbf{x}}_i$ is the $i$-th row of $\tilde{\mathbf{X}}$. The vector of coefficients $\mathbf{a}_1$ is found by maximizing the sample variance of the new variable:

$$\text{Var}(z_{11}, z_{21}, \ldots, z_{n1}) = \frac{1}{n-1}\sum_{i=1}^{n}(z_{i1} - \bar{z}_1)^T(z_{i1} - \bar{z}_1) = \mathbf{a}_1^T\mathbf{S}\mathbf{a}_1 \tag{2}$$

subject to the constraint $\mathbf{a}_1^T\mathbf{a}_1 = 1$ to ensure finite $\mathbf{a}_1$. This problem can be reformulated as a Lagrange function:

$$\mathcal{L}(\mathbf{a}_1, \lambda_1) = \mathbf{a}_1^T\mathbf{S}\mathbf{a}_1 - \lambda_1(\mathbf{a}_1^T\mathbf{a}_1 - 1) \tag{3}$$

where $\lambda_1$ is a Lagrangian multiplier. The stationary solutions are calculated by the partial derivatives with regards to $\mathbf{a}_1$:

$$\frac{\partial}{\partial\mathbf{a}_1}\left(\mathbf{a}_1^T\mathbf{S}\mathbf{a}_1 - \lambda_1(\mathbf{a}_1^T\mathbf{a}_1 - 1)\right) = 0 \tag{4}$$

This yields $\mathbf{S}\mathbf{a}_1 = \lambda_1\mathbf{a}_1$. We can conclude that the variation is maximized when the vector of coefficients $\mathbf{a}_1$ is the eigenvector of the sample covariance matrix corresponding to the largest eigenvalue:

$$\text{Var}(z_{11}, z_{21}, \ldots, z_{n1}) = \mathbf{a}_1^T\mathbf{S}\mathbf{a}_1 = \lambda_1 \tag{5}$$

The next principal components are calculated analogously. In general, the optimal linear projection of the mean-subtracted dataset $\tilde{\mathbf{X}}$ along $p$ principal components is calculated using the transformation:

$$\mathbf{Z} = \tilde{\mathbf{X}}\mathbf{A} \tag{6}$$

where $\mathbf{Z}$ is an $n \times p$ matrix of $p$ principal components of $n$ different observations and $\mathbf{A}$ is a $d \times p$ matrix of $p$ eigenvectors.

In modern applications, PCA is often calculated by solving SVD problem (Stewart, 1993). SVD can be solved in different ways. We use the randomized method (Halko et al., 2011) to solve SVD independently of $d$, making it much more efficient, especially for high-dimensional data. In order to do so, we form an $n \times p$ matrix $\mathbf{Y}$ by multiplying alternately with $\tilde{\mathbf{X}}$ and $\tilde{\mathbf{X}}^T$ using the following equation:

$$\mathbf{Y} = (\tilde{\mathbf{X}}\tilde{\mathbf{X}}^T)^q\tilde{\mathbf{X}}\mathbf{\Omega} \tag{7}$$

where $\mathbf{\Omega}$ is a generated $d \times p$ Gaussian random matrix and $q$ is an exponent. Then, we construct an $n \times p$ matrix $\mathbf{Q}$ such that its columns form an orthonormal basis for the range of $\mathbf{Y}$. After the construction of $\mathbf{Q}$, we calculate the $p \times d$ matrix $\mathbf{B}$:

$$\mathbf{B} = \mathbf{Q}^T\tilde{\mathbf{X}} \tag{8}$$

$\mathbf{B}$ is a small matrix as often $p \ll d$. The SVD of $\mathbf{B}$ is calculated by:

$$\mathbf{B} = \hat{\mathbf{U}}\mathbf{\Sigma}\mathbf{V}^T \tag{9}$$

where $\mathbf{\Sigma}$ is a $p \times d$ diagonal matrix and $\hat{\mathbf{U}}$ and $\mathbf{V}$ are $p \times p$ and $(d \times d)$ orthogonal matrices respectively. Finally, we calculate the $n \times p$ matrix $\mathbf{U}$ by:

$$\mathbf{U} = \mathbf{Q}\hat{\mathbf{U}} \tag{10}$$

$\mathbf{U}$ contains the $p$ principal components of the data.

## 4 Randomized PCA Forest

In this paper, we propose a novel unsupervised outlier detection method based on RPCA forests (Rajabinasab et al., 2024). The performance of the RPCA forest in approximate KNN search motivates its adaptation for the outlier detection task. RPCA forest is an ensemble of RPCA trees employing randomized PCA (Halko et al., 2011) in their design. The construction of a tree is done by the following process: First, in the root of the tree, the entirety of data points are contained. Then, randomized PCA is employed to project the data points into a lower-dimensional subspace in which most of the informative value of the original feature space is preserved. Randomized PCA offers a faster and more efficient alternative compared to the traditional PCA by using a randomized Singular Value Decomposition (SVD) solver, as its complexity is tied to the number of principal components being calculated and not the dimensionality of the data (Halko et al., 2011). A unique splitting criterion is then generated to divide the data points into the left and the right child by their coordinates in the principal components space. This process continues until the stopping criterion is met. The stopping criterion is based on the (approximate) neighborhood size (e.g., maximum size of the tree leaves). Outlier detection using RPCA forest stands on the idea that, first, the outliers are more likely to appear inside the leaf nodes which are located in the lower depths of the tree, and second, their average distance in the original feature space from the other data points in the same leaf node is larger compared to the inliers. In the remainder of this section, we discuss each part of the outlier detection process separately.

### 4.1 RPCA Tree

The RPCA tree is the backbone of RPCA forest which constructs the forest by forming an ensemble. Alg. 0 describes the process of fitting an RPCA tree. Let *root* be the root node of the RPCA tree $\mathcal{T}$ which contains all of the data points. Let $p$ be the number of principal components used in $\mathcal{T}$ and $k$ be the (approximate) neighborhood size. $k$ is used for the stopping criterion by setting the maximum leaf size to be less than or equal to $k$. As there might be fewer data points in the leaf node, the neighborhood size is approximate. The fitting process of the tree starts with all the data points in the root node. Then, data points are projected into $p$ corresponding principal components using randomized PCA to form a lower-dimensional subspace in which most of the informative value of the features is preserved. A splitting point is sampled for each principal component from a Laplace distribution over the subspace. Then, by employing majority voting, a decision is made to assign data points to the left or right child. The fitting process continues for each child node until every branch reaches the stopping criterion, which is the approximate neighborhood size or the maximum size for a node to be considered a leaf. An illustration of a Laplace distribution is shown in Fig. 1. It is evident that using a Laplace distribution is a good choice for finding the splitting criteria as it assigns

---

**Algorithm 1** RPCA Tree

---

**Input:** The dataset $D$, number of principal components $p$, and the neighborhood size $k$
**Output:** Fitted RPCA tree $\mathcal{T}$
 1: Let $root_D$ be the root node of $\mathcal{T}$;
 2: Initialize the set of working nodes $\mathcal{W} \leftarrow \{root_D\}$;
 3: **while** $\mathcal{W}$ is not empty **do**
 4:     Pick $n$ and set $\mathcal{W} \leftarrow \mathcal{W} \setminus \{n\}$;
 5:     **if** $|n| \leq k$ **then**
 6:       Skip to the next round;
 7:     **end if**
 8:     Calculate $E$, set of $p$ eigenvectors for data points in $n$;
 9:     $PC \leftarrow project(\mathbf{x}, E), \forall \mathbf{x} \in n$
10:     **for** $i = 1$ to $p$ **do**
11:       $s_i \leftarrow laplace(mean(PC_i), std(PC_i))$;
12:     **end for**
13:     **for** each data point $\mathbf{x} \in n$ **do**
14:       Initialize $c_{left}, c_{right} \leftarrow 0$;
15:       **for** $i = 1$ to $p$ **do**
16:         **if** $PC_i < s_i$ **then**
17:           $c_{left} = c_{left} + 1$;
18:         **else**
19:           $c_{right} = c_{right} + 1$;
20:         **end if**
21:       **end for**
22:     **end for**
23:     **for** each data point $\mathbf{x} \in n$ **do**
24:       **for** $i = 1$ to $p$ **do**
25:         **if** $c_{left} > c_{right}$ **then**
26:           $n_{left} \leftarrow \{\mathbf{x}\}$;
27:         **else**
28:           $n_{right} \leftarrow \{\mathbf{x}\}$;
29:         **end if**
30:       **end for**
31:     **end for**
32:     $\mathcal{W} \leftarrow \mathcal{W} \cup \{n_{left}, n_{right}\}$;
33: **end while**
34: **return**$(\mathcal{T})$;

---

higher probability to the values close to the mean, leading to balanced splits for different datasets and faster fitting process.

We can ensemble the desired number of RPCA trees to form an RPCA forest. The RPCA forest demonstrated a promising performance in approximate KNN search and it was observed that with increasing the number of the trees in the forest, one can improve the performance of KNN search (Rajabinasab et al., 2024). We follow the same principle here, but we define an outlier score based on the placement of a point in the tree and its distance from other points to perform the unsupervised outlier detection task.

### 4.2 Outlier Score

We can utilize an RPCA forest for unsupervised outlier detection by defining an outliers score. Let $T$ be the set of RPCA trees $\mathcal{T}$ and $\mathbf{q}$ the query point for outlier detection. Let $L_{\mathbf{q}}$ be the leaf node of $\mathcal{T}$ that includes $\mathbf{q}$ if we traverse the tree using $\mathbf{q}$ based on the projections and splitting criterion calculated during the fitting process. The depth-based outlierness of $\mathbf{q}$ is defined as

$$P(\mathbf{q}) = \frac{\sum_{\mathcal{T} \in T} \left(1 - \frac{\text{Depth}(L_{\mathbf{q}})}{\text{Depth}(\mathcal{T})}\right)}{|T|} \tag{11}$$

where $\text{Depth}(L_{\mathbf{q}})$ is the depth of the leaf node in which $\mathbf{q}$ lies and $\text{Depth}(\mathcal{T})$ is the depth of the tree (maximum depth among all of the leaf nodes).

The depth-based outlierness criterion is based on the idea that outliers are more likely to reach a leaf node earlier (higher up) in the tree as they are different from other observations, i.e., their depth will be small.

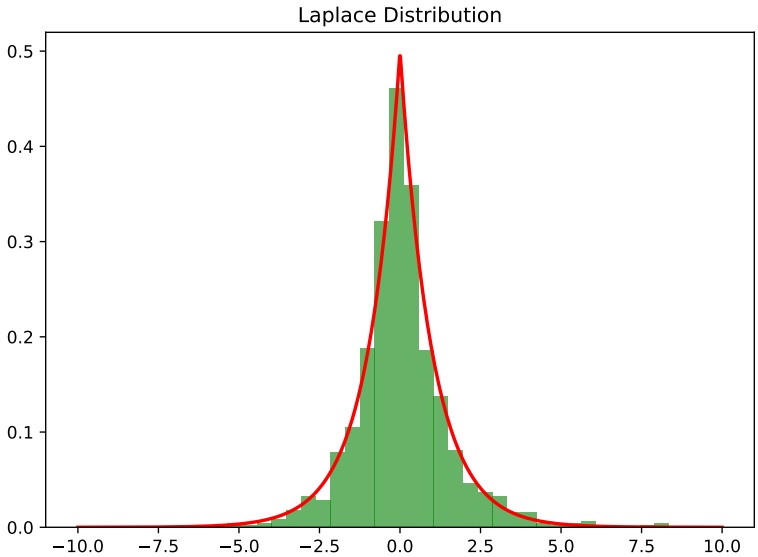

Figure 1: An illustration of the Laplace distribution.

The benefit of this criterion is the comparability across different datasets and settings. However, in order to have a precise outlierness criterion for the unsupervised outlier detection task, $k$ must be set to 1. If $k$ is set to a larger value, data points with different coordinates will share the same value. On the other hand, setting $k$ to 1 poses another limitation. As we are using PCA in each node and the number of principal components being calculated can not be larger than $\min(n_{\text{instances}}, n_{\text{features}})$, the upper bound of 2 will be imposed on $p$. These limitations highlight the need for a more robust and granular outlierness criterion.

To address the shortcoming of the depth-based outlierness criterion, we introduce a more precise and granular criterion for outlierness, the mean of the distances of a query point to every point in its corresponding leaf:

$$\mu_{dist}(\mathbf{q}) = \frac{\sum_{\mathcal{T} \in T} \frac{\sum_{\mathbf{x} \in L_{\mathbf{q}}} d(\mathbf{q}, \mathbf{x})}{|L_{\mathbf{q}}|}}{|T|} \tag{12}$$

where $d(\cdot, \cdot)$ is the Euclidean distance.

The final outlier score for a point $\mathbf{q}$ is then the mean distance of $\mathbf{q}$ within its leaf node weighted by its depth-based criterion:

$$RPCAForest_{Score}(\mathbf{q}) = P(\mathbf{q})\mu_{dist}(\mathbf{q}) \tag{13}$$

By incorporating the distance in the original feature space and the depth-based outlierness in the calculation of the outlier score, we make sure that even if, based on the representation in the principal component space, there are some inliers mixed with the outlier in the same leaf node, they are well separated as the mean of the distances to the data points in the leaf is expected to be higher for outliers compared to the inliers.

Unsupervised outlier detection is done by using a contamination parameter which denotes the percentage of outliers expected to exist in the dataset. Then, based on the contamination, a cutoff value is chosen as a threshold for a point to be an outlier. The RPCA Forest's outlier detection process is presented in Alg. 2. The fitting the RPCA Forest follows the original procedure by Rajabinasab et al. (2024).

Theoretically we can expect, from RPCA forest's perspective, that points which are more easily separated from others based on their representation using the first $p$ principal components and show a high average distance in the original space to the points which are similar to them based on that representation, are more likely to be considered outliers. If we follow a point from the root to the leaf, we observe:

- The global outliers, which are different from most of the observations in the dataset, are expected to be separated more quickly, reaching a leaf node in a low depth.

---

**Algorithm 2** RPCA Forest Outlier Detection

---

**Input:** Dataset $D$, fitted RPCA Forest $T = \{\mathcal{T}_1, \ldots, \mathcal{T}_{|T|}\}$, and contamination rate $\alpha$
**Output:** Outlier scores $S$, Label set $Y$, Threshold $c$
 1: Initialize $S \leftarrow []$
 2: **for** each point $\mathbf{x} \in D$ **do**
 3:     Initialize $P_{\text{total}} \leftarrow 0$
 4:     Initialize $\mu_{\text{total}} \leftarrow 0$
 5:     **for** each tree $\mathcal{T} \in T$ **do**
 6:         $L_{\mathbf{x}} \leftarrow FindLeaf(\mathbf{x}, \mathcal{T})$
 7:         $P_{\text{tree}} \leftarrow 1 - (\text{Depth}(L_{\mathbf{x}}) / \text{Depth}(\mathcal{T}))$
 8:         $P_{\text{total}} \leftarrow P_{\text{total}} + P_{\text{tree}}$
 9:         $\mu_{\text{leaf}} \leftarrow Average(\{d(\mathbf{x}, \mathbf{p}) \mid \mathbf{p} \in L_{\mathbf{x}}\})$
10:         $\mu_{\text{total}} \leftarrow \mu_{\text{total}} + \mu_{\text{leaf}}$
11:     **end for**
12:     $P(\mathbf{x}) \leftarrow P_{\text{total}}/|T|$
13:     $\mu_{dist}(\mathbf{x}) \leftarrow \mu_{\text{total}}/|T|$
14:     $Score(\mathbf{x}) \leftarrow P(\mathbf{x}) \times \mu_{dist}(\mathbf{x})$
15:     $S.append(Score(\mathbf{x}))$
16: **end for**
17: $c \leftarrow Quantile(S, 1 - \alpha)$
18: Initialize $Y \leftarrow []$
19: **for** each score $s \in S$ **do**
20:     **if** $s > c$ **then**
21:         $Y.append(\text{'Outlier'})$
22:     **else**
23:         $Y.append(\text{'Inlier'})$
24:     **end if**
25: **end for**
26: **return** $(S, Y, c)$

---

- PCA algorithm applied on the data points in each node, becomes theoretically more local, working with a group of points with increasing similarity as we go towards the leaves. The points which are different from every other point, are expected to be separated from others in early stages.

- No matter how similar the points are in the principal component space, they will still be effectively divided into two, if the size of the node is greater than $k$. Even if the principal component representation is strongly different from the original, we will still get a high value for outlier score.

Based on this observation, we expect RPCA Forest to be effective in detecting both local and global outliers (Schubert et al., 2014), beginning with separating and discriminating global outliers, and continuing with detecting local outliers in nodes further down the tree structure.

### 4.3 Computational Complexity

The computational complexity of constructing an RPCA tree is determined by the number of data points $n$ and the number of principal components $p$. The construction process entails recursively partitioning the dataset using randomized PCA. The expected computational complexity for building an RPCA tree with $n$ data points is $O(\log^2(n)p + \log(n)p^3)$. For the search phase, the average time complexity is $O(\log(n))$, corresponding to the traversal from the root to a leaf node. However, the search cost is also influenced by the approximate neighborhood size, which sets the maximum number of points stored in a leaf node. A larger neighborhood size reduces the tree's depth, potentially decreasing the traversal time but increasing the computational cost of processing the leaf node's contents. Calculating the distance-based part of the outlier score has the complexity of $O(kp)$. This computational cost becomes insignificant when $k$ and $p$ are small (e.g., $k = 10$ and $p = 1$). On the other hand, it takes more time for the number of data points in a node to become less than $k$. This trade-off enables the tree's structure to be tuned based on specific application requirements. Calculating the depth-based part of the outlier score is computationally insignificant: The depth of each node can be calculated and stored by a simple sum operation during the fitting process and the maximum depth can also be determined equally efficiently.

Furthermore, RPCA trees are well-suited for parallelization. Since each tree is constructed independently, multiple trees can be built concurrently, leveraging parallel computing resources to significantly reduce the overall construction time. This property is particularly advantageous in multi-tree scenarios, such as those encountered in randomized forest-based methods.

### 4.4   RPCA Forest vs. iForest

As iForest (Liu et al., 2008) is a very popular tree-based method for unsupervised outlier detection, let us briefly discuss its similarities and differences to RPCA Forest. In iForest, the outlier detection process is based on the assumption that outliers are "few and different". An iForest consists of several isolation trees which select one feature randomly and generate random splitting points to partition the data. This process continues until every data point is "isolated" in a leaf. Based on the assumption that the outliers are different, they are expected to be isolated more easily and in lower depth leaf nodes. Hence, the depth in which the data point lies can be directly utilized for outlier detection. An outlier score can be derived from the depth of a point in the tree and the maximum depth of the tree. The outlier scores can be averaged over the trees in a forest to provide the forest's decision.

RPCA Forest uses a combination of the depths and the average distance to the other points in the leaf to calculate the outlier scores. It does not tend to isolate the data points and it can operate on leaf nodes with an arbitrary number of points lying in them. On the other hand, RPCA Forest does not pick features randomly and utilizes RPCA to operate in a lower-dimensional informative subspace where the subspace dimensionality can be above one. In conclusion, the only similarity of the proposed method and iForest is the utilization of the tree structure to conduct the unsupervised outlier detection task.

## 5   Experiments and Evaluation

For the experimental evaluation, we use 22 different datasets selected from the repository of a large comparison study (Campos et al., 2016). The datasets are detailed in Table 1. The Lymphography dataset is excluded from the experiments as it has only 3 features and is also considered to be an extremely easy dataset for outlier detection (Campos et al., 2016). We use the Area Under the Curve (AUC) of the receiver operating characteristic (ROC) to investigate the overall performance of the compared methods:

$$\text{AUC} = \int_0^1 \text{TPR}(\text{FPR}) \, d\,\text{FPR} \tag{14}$$

where TPR and FPR are True Positive Rate and False Positive Rate, respectively, and ROC plots TPR as a function of FPR. This is a robust, threshold-independent metric that effectively measures a model's ability to distinguish outliers from inliers across all possible decision boundaries, ensuring reliable evaluation even in the presence of highly imbalanced datasets where outliers are rare.

### 5.1   The Effect of Forest Size

To investigate the effect of forest size in the outlier detection performance, we select 3 different datasets, Arrhythmia, Glass, and WDBC, and we calculate the AUC values for up to 100 trees. Experiments on each forest size are done with 5 different selections of trees for the forest to provide more robust results. Fig. 2 depicts the results of this experiment. Overall, the forest for all datasets stabilizes towards its optimal value with approximately 20 trees. Arrhythmia is more unstable as it has the highest number of features and more features than data points. The analysis shown is for $p = 1$ and $max\_size$ of the leaf $k = 10$, however, we observed that the figure is fairly similar for other choices of parameters.

### 5.2   Hyperparameter Investigation

We conduct experiments with different choices for parameters. For $max\_size$ of the leaf, or the approximate neighborhood size, which is denoted by $k$, we use 10 and 20. For the number of principal components $p$, we use 1 and 5. Clearly, there is an infinite number of possibilities for the choice of parameters. As it is not

Table 1: Overview of the datasets used in the experiments.

| Key | Dataset | Instances (#) | Attributes (#) | Outliers (%) |
|---|---|---|---|---|
| D1 | ALOI | 49,534 | 27 | 3.04 |
| D2 | Annthyroid | 6,729 | 21 | 1.99 |
| D3 | Arrhythmia | 248 | 259 | 1.61 |
| D4 | Cardiotocography | 1,681 | 21 | 1.96 |
| D5 | Glass | 214 | 7 | 4.21 |
| D6 | HeartDisease | 153 | 13 | 1.96 |
| D7 | Hepatitis | 70 | 19 | 4.29 |
| D8 | InternetAds | 1,630 | 1555 | 1.96 |
| D9 | Ionosphere | 351 | 32 | 35.9 |
| D10 | KDDCup99 | 48,113 | 40 | 0.42 |
| D11 | PageBlocks | 4,982 | 10 | 1.99 |
| D12 | Parkinson | 50 | 22 | 4.0 |
| D13 | PenDigits | 9,868 | 16 | 0.2 |
| D14 | Pima | 510 | 8 | 1.96 |
| D15 | Shuttle | 1,013 | 9 | 1.28 |
| D16 | SpamBase | 2,579 | 57 | 1.98 |
| D17 | Stamps | 315 | 9 | 1.9 |
| D18 | WBC | 223 | 9 | 4.48 |
| D19 | WDBC | 367 | 30 | 2.72 |
| D20 | WPBC | 198 | 33 | 23.74 |
| D21 | Waveform | 3,443 | 21 | 2.9 |
| D22 | Wilt | 4,655 | 5 | 2.0 |

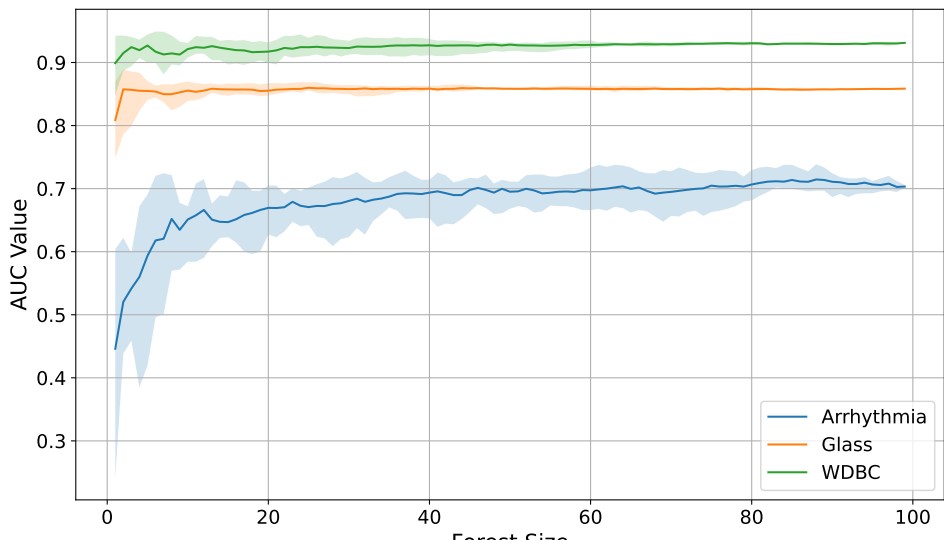

Figure 2: The effect of forest size on the performance of RPCA forest.

feasible to conduct experiments based on all of the possible choices, we use the combination of these values to investigate their effect on performance of the proposed method. All models consist of 100 trees.

Fig. 3 shows the performance of the models with different hyperparameter pairs. In most cases the models with $k = 10$ perform better. We conjecture this is mainly due to the higher accuracy in the outlier scores as this leaf size makes sure that there are enough inliers included in each leaf node. For the number of principal components $p$, it is not obvious which parameter choice is better. It seems that in general, $p = 1$ performs better, but still $p = 5$ is the better choice in some cases.

To investigate the effect of $p$ further, we plot the explained variance ratio by the principal components for $p = 1$ and $p = 5$ for each dataset in Fig. 4. We observe that, for datasets in which the difference in the explained variance ratio for $p = 1$ and $p = 5$ is small and $p = 1$ contributes to a high proportion of explained variance in $p = 5$, the performance of the models with $p = 1$ is better or competitive. When the difference is larger, $p = 5$ performs marginally better. This analysis can be a good way to find the optimal value for $p$. However, in most cases, the differences are small, and either choice might be good enough.

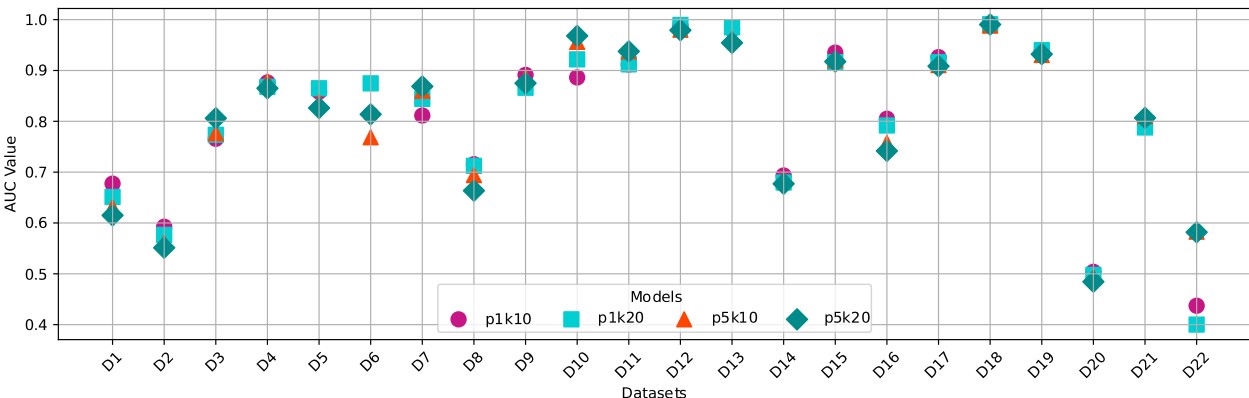

Figure 3: The performance of RPCA forest using different hyperparameter combinations.

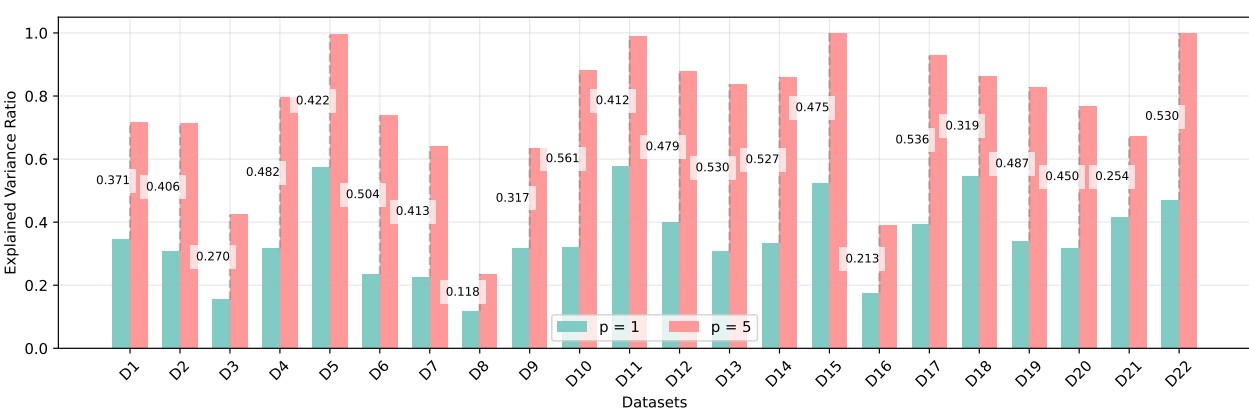

Figure 4: The investigation of the amount of explained variance ratio using $p = 1$ and $p = 5$. The values on the figure indicate the absolute difference in the amount of explained variance ratio between two cases.

## 5.3 Comparative Analysis

We present the experimental results across the 22 datasets, comparing our proposed method using different hyperparameters with other methods including KNN (Ramaswamy et al., 2000), KNNW (Angiulli & Pizzuti, 2002), LOF (Breunig et al., 2000), SLOF (Schubert et al., 2014), ODIN (Hautamaki et al., 2004), DAO (Anderberg et al., 2024), and iForest (Liu et al., 2008). Note that most methods share the notion of neighborhood size as a parameter $k$, only iForest does not have such a parameter. The experimental results for $k = 10$ and $k = 20$ are presented in Fig. 5 and 6. For the proposed method and iForest, a box plot is shown over 10 different runs to account for their random nature. The experimental results show a diverse figure for the performance of different models on different datasets. The proposed method shows the best or second-best results in many cases both for $k = 10$ and $k = 20$. In others, it shows competitive figures. The box plots also indicate that despite the randomness in the underlying process of the proposed method, it shows to be fairly stable in conducting the unsupervised outlier detection task. It is noteworthy that hyperparameter tuning is not done on the proposed method and only two different values for $p$ are used. The results for other methods are available from the benchmark study by Campos et al. (2016). We conducted the experiments for DAO (Anderberg et al., 2024) based on their implementation.

There are some datasets, on which the proposed method shows rather poor performance. These datasets include ALOI, Annthyroid, InternetAds, and Wilt. Based on the theoretical understanding of the method's principles, and the empirical results, we briefly discuss possible reasons for a poor performance observed

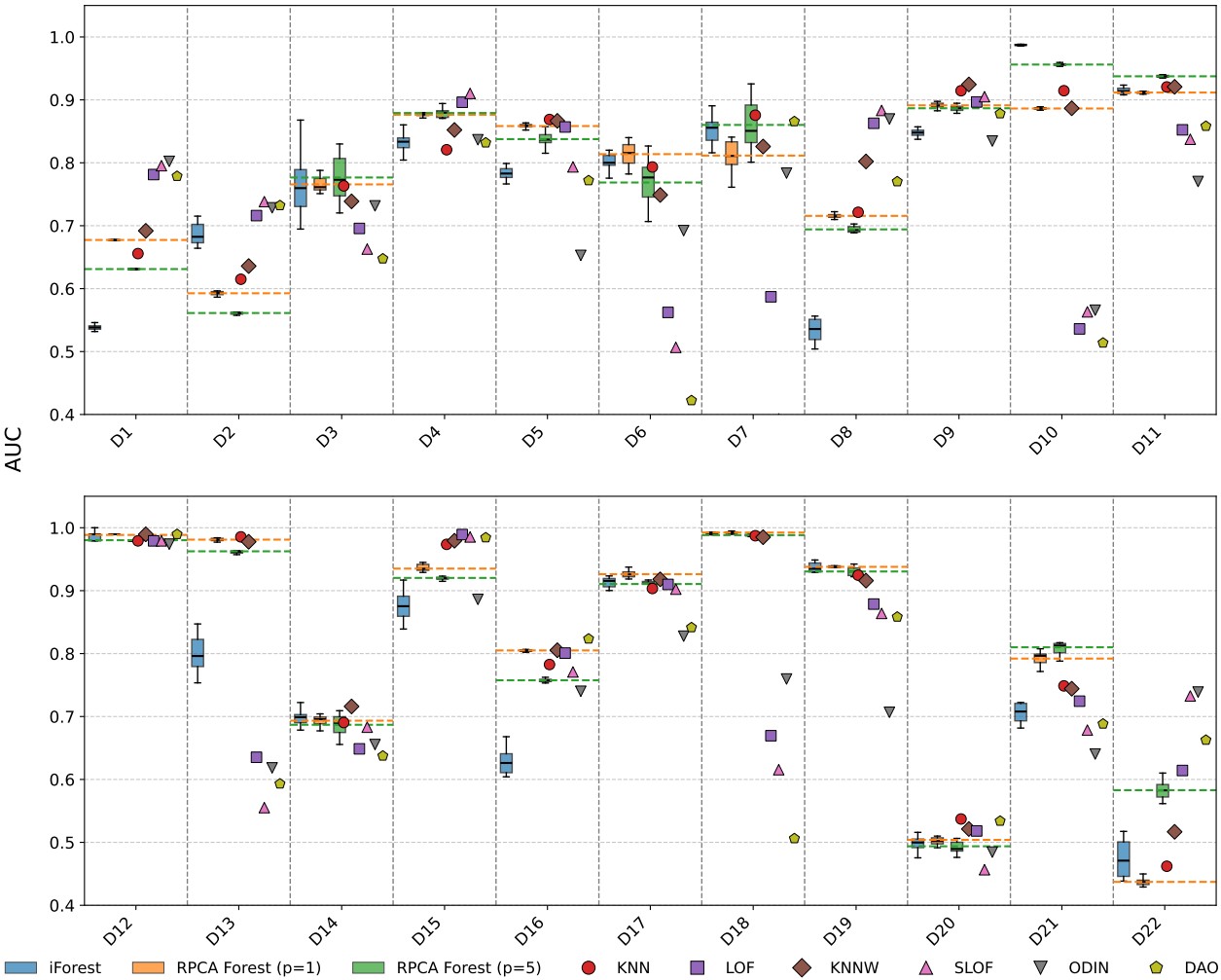

Figure 5: AUC values of the proposed method and the competitors for $k = 10$. Dashed lines are used to indicate the mean AUC value for the proposed method.

in these cases. For InternetAds, as observed later in the ablation study, the depth-based part of the score significantly fails. This is due to the high dimensionality of this dataset ($d = 1555$), which renders a few principal components ($p = 1$ or $p = 5$) insufficient for successfully isolating outliers in low depths. Annthyroid and Wilt also suffer from the same shortcoming, just from a different perspective. In both datasets, the outliers are very similar to inliers. In these cases, more information is required to successfully discriminate outliers from inliers. Looking at Fig. 3, it is evident, especially for the Wilt dataset where $p$ becomes equal to $d$, that outlier detection performance becomes much better when $p = 5$ compared to the case with $p = 1$. For Annthyroid and InternetAds, it is less obvious as more principal components are still required to avoid under-representing the original features. For ALOI, the poor performance is due to the case of local vs. global outliers. ALOI is a dataset with approximately 50k instances and 3% outliers. In ALOI, there are 27 numerical features, corresponding to the Hue-Saturation-Brightness (HSB) histogram. This feature extraction approach makes images labeled as outliers very close to the inlier instances. This is the reason that methods such as LOF which rely solely on local structures provide a better result on ALOI compared to the proposed method, but a method like iForest, fails even more significantly.

To have a better overview of the comparison of the proposed method with the competitors, we present the critical difference diagram for the cases $k = 10$ and $k = 20$. We use the best AUC value of the $p = 1$ and

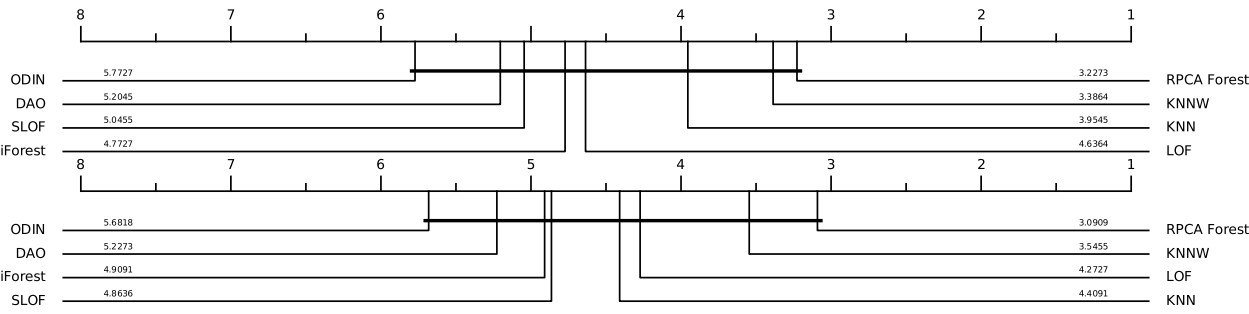

Figure 6: AUC values of the proposed method and the competitors for $k = 20$. Dashed lines are used to indicate the mean AUC value for the proposed method.

Figure 7: The critical difference diagram based on AUC. $k = 10$ on the top and $k = 20$ on the bottom. The methods to the right show a better average ranking across all datasets.

$p = 5$ cases for the RPCA Forest in this diagram. The critical difference diagram (Fig. 7) shows the result of the rank analysis and orders method based on their performance from right to left. The diagram follows the methodology of Demšar (2006). The Friedman test is used to detect significant differences in performance,

followed by a post-hoc Wilcoxon signed-rank test with Holm's correction to identify pairwise differences. Based on empirical observations and statistical analysis of the CD diagrams, it is evident that the proposed method shows better (albeit not significant overall) performance figures than the competitors.

### 5.4 Robustness Analysis

To demonstrate how robust the proposed method is, we conduct a broader comparison with the competing methods. In this comparison, 100 different hyperparameters of all of the competing methods are tested to generate the box plots for AUC values. These results are extracted from the work done by Campos et al. (2016) for the methods tested there. We did the same hyperparameter tuning for DAO (Anderberg et al., 2024). For the proposed method, we only keep the variant with $p = 1$ as the baseline. We conducted experiments only on two hyperparameters to see how robust the proposed algorithm is. In real world scenarios, as we do not have access to the labels, we cannot realistically conduct hyperparameter tuning. Hence, an algorithm which performs well in general is preferred. Fig. 8 shows the results of this experiment.

We can observe that despite exploring 100 different parameters for all the competing methods (except iForest, for which this is not applicable), the proposed method with only two different tested parameters shows either improved or competitive performance. Only in a few cases, a weaker performance in terms of AUC is observed by the proposed method. The experimental results indicate that the proposed method is highly robust and can be effectively used for outlier detection without the need of an exhaustive hyperparameter tuning phase.

### 5.5 Ablation Study

The outlier score proposed for RPCA Forest is a combination of depth-based and distance-based approaches to outlier detection. Despite the acceptable performance demonstrated by the algorithm using this outlier score, it is important to shed light on the importance of each part and why combining these scores makes sense. In order to achieve this, we repeat our experiments using the depth-based and the distance-based part of the outlier score individually to shed light on their standalone performance in outlier detection. In Fig 9, the results of the ablation study for different choices of $p$ and $k$ on all 22 datasets are presented.

As demonstrated in Fig 9, the final outliers score outperforms the depth-based part of the score, and appears to be very competitively performing against the distance-based part. In order to get a clearer overview of the results, we use CD diagrams for each case to demonstrate how the scores perform more transparently. CD diagrams are presented in Fig 10.

It is evident that in three out of the four cases, the final outlier score and the distance-based score perform exactly the same in terms of ranking statistics, outperforming the depth-based score by a high margin. However with $p = 5$ and $k = 10$, we observe that, despite still being in the same statistical range, the combined outlier score performs better than the two components alone. This indicates that combining scores to calculate the final outlier score is a good choice for the robustness of the algorithm in outlier detection. As incorporating the depth does not introduce any additional computational cost, including it in the final outlier score is the natural choice to increase overall stability.

### 5.6 Comparison with Isolation Forest

We observed that iForest (Liu et al., 2008) is the fastest and most efficient method among all the competitors, however in terms of AUC, it performed mediocre. Nevertheless, its performance is acceptable and promising in general. It is also the only competitor which uses a tree-based structure for outlier detection. Thus, we compare the proposed method with iForest in terms of convergence rate. In other words, we investigate how many trees are required for iForest to reach optimal performance compared to the proposed method. For this experiment, we test the performance of the proposed method and iForest with using different numbers of trees. We use 100 as the maximum size for the forest. The datasets we use for this experiment are Arrhytmia, WDBC, and Waveform. The result of this experiment are depicted in Fig. 11.

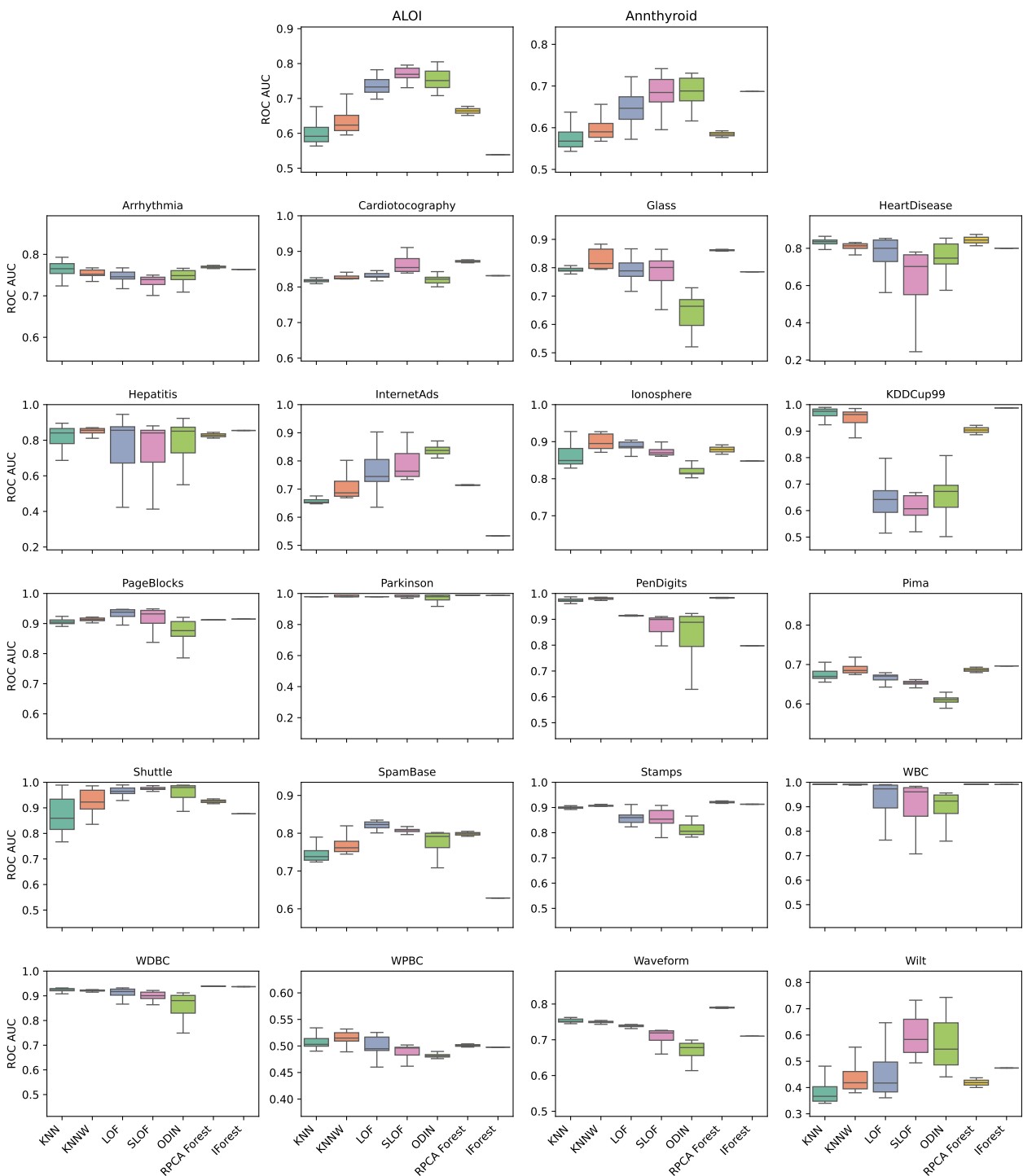

Figure 8: Robustness analysis. The box plots show the observed AUC values in the evaluation of the competitors with 100 different hyperparameters. Our method was only tested using two different choices.

It is evident that in general, RPCA Forest shows a better performance compared to iForest when few trees are used. This phenomenon is more clear for the Arrhythmia dataset, where the dimensionality is high. In general, as the proposed method uses PCA to perform outlier detection based on an informative subspace, it

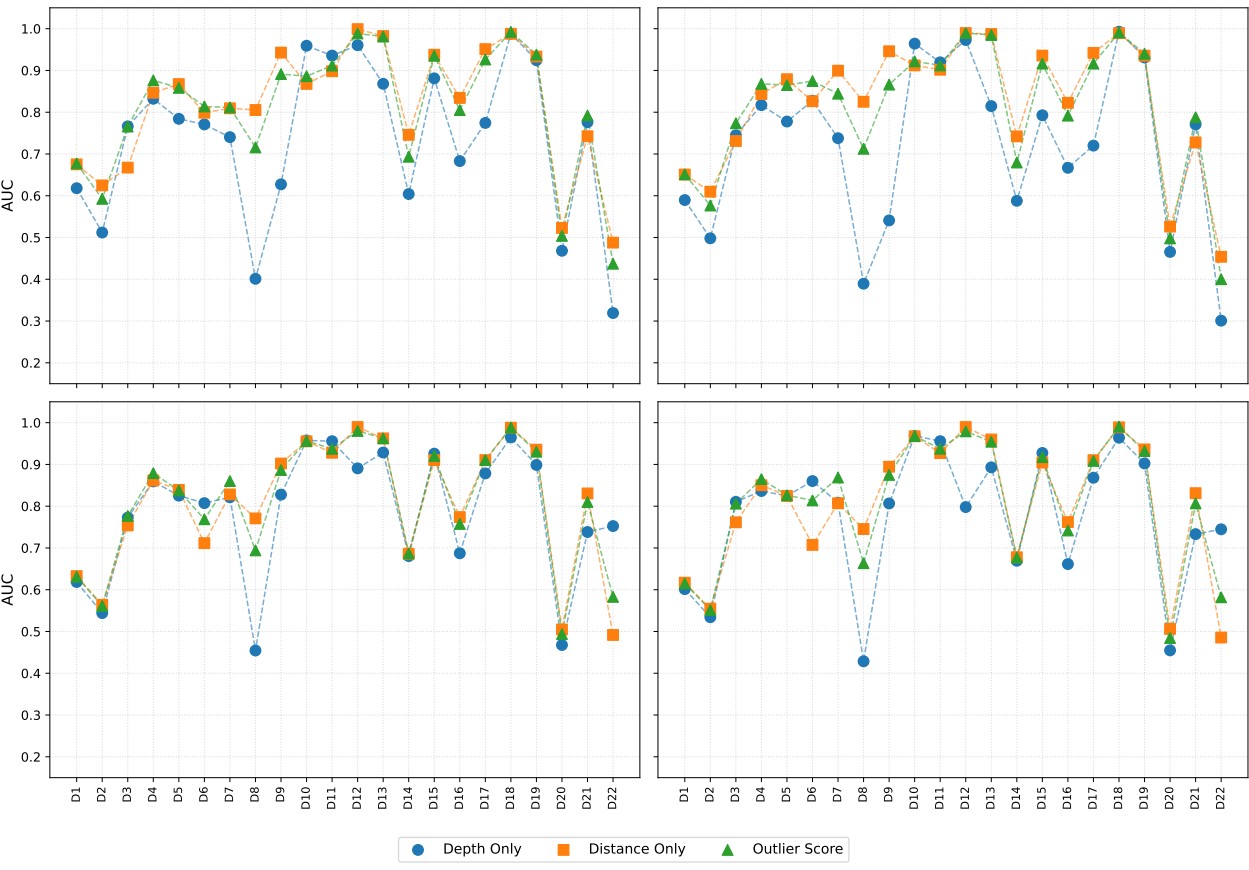

Figure 9: Investigation of the depth-based and distance-based parts of the outlier score against the outlier score itself for different values of $p$ and $k$ across 22 datasets. $p = 1$ at the top and $p = 5$ at the bottom. $k = 10$ on the left, and $k = 20$ on the right.

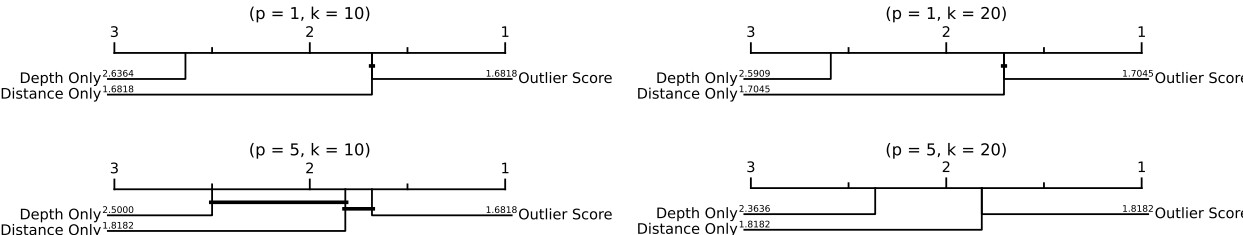

Figure 10: Critical Difference diagrams for different $p$ and $k$ values for different outlier score criteria.

works better on high-dimensional and complex datasets compared to iForest which selects features randomly to fit the trees. On the other hand, the outlierness criterion used in RPCA Forest is more advanced and granular compared to the iForest, which leads to better performance. However, iForest still remains a remarkable solution for outlier detection as its underlying mechanism is considerably fast.

## 5.7 Efficiency

Most outlier detection methods rely on the $k$ nearest neighbors for each point, which makes them computationally expensive (Kirner et al., 2017). Every competitor in this study except iForest is involved with finding nearest neighbors as a part of their outlier detection process. iForest is the most efficient among all of the methods and it is the only method alongside with RPCA Forest that is based on a tree structure.

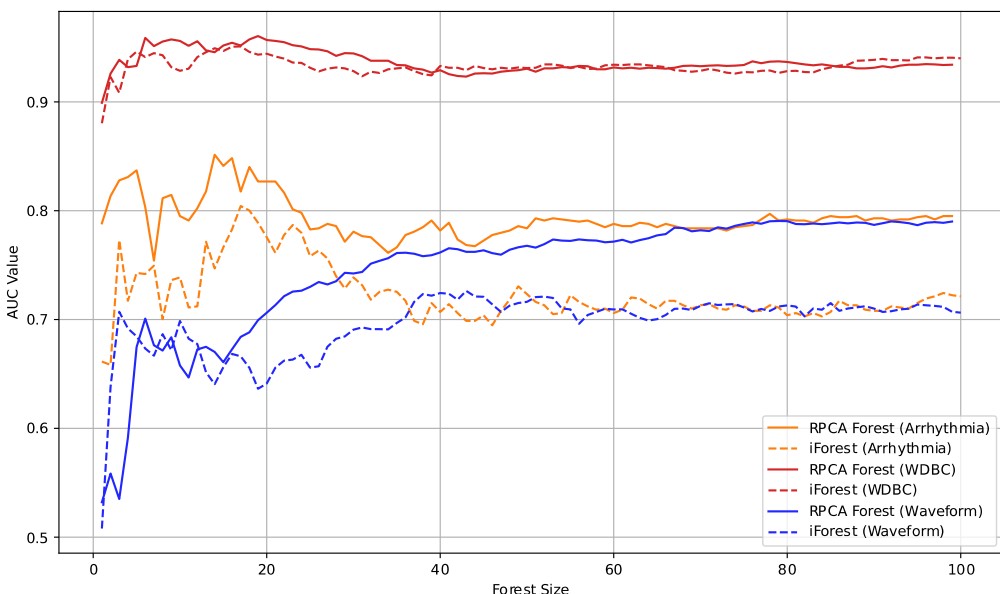

Figure 11: Comparison of the forest size effect on the performance of the proposed method with iForest.

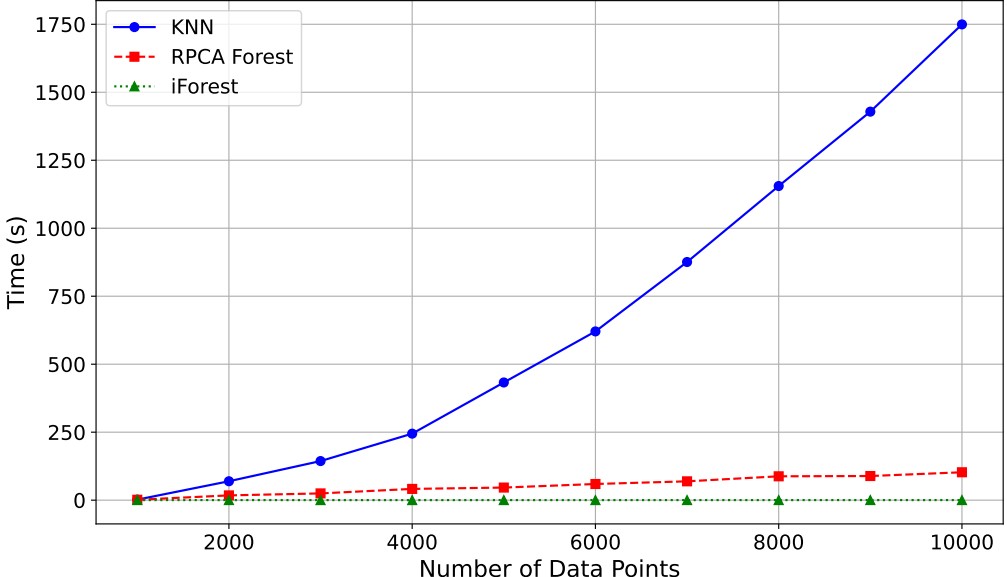

Figure 12: Comparison of the effect of dataset size on the running time of RPCA Forest, KNN, and iForest.

To further investigate the efficiency of RPCA Forest, we conduct time complexity analysis based on dummy datasets with different sizes and fixed dimensionality (100). We compare the running time of RPCA Forest with KNN as the backbone of several competitors, and iForest. The results in Fig. 12, show that as the number of data points increases, the time required for outlier detection grows rapidly, while RPCA Forest maintains an efficient runtime, demonstrating its ability in conducting the outlier detection task on big data. As expected, the fastest method is iForest, but its efficiency margin is insignificant, considering that its performance was consistently not among the best.

## 6 Conclusion and Future Work

In this paper, we introduced a novel outlier detection method leveraging RPCA forest that showed robust performance in approximate KNN search. We conducted extensive experiments across multiple datasets, comparing its performance against several established outlier detection methods. The experimental results demonstrate that the proposed method shows improved performance compared to the competitors using the same parameter choices. Using optimal parameters for the competing methods while only two hyperparameter settings for the proposed method, it still demonstrated competitive performance. The results underscore the method's robustness, achieving improvement in performance on several datasets and competitive results on others, even without exhaustive hyperparameter optimization. The proposed method's computational efficiency and its ability to perform effectively across diverse datasets and experimental settings are its key strengths.

Despite the improvements in performance figures using RPCA forest, there might be some theoretical limitations to the methodology it uses for outlier detection. As discussed in Section 4, we expect RPCA forest to be effective in detecting both global and local outliers. However, in the case of mini-clusters of anomalous points, or a dense area of outliers in the dataset, the distance-based part of the outlier score might fail, leading to false predictions in cases such as collective anomalies (Chandola et al., 2009). The proposed method shares this limitation with many fundamental and classic methods for outlier detection. Specialized methods for dealing with clustered outliers are required, when such a scenario is expected (Jiang et al., 2022).

For future research, we can develop additional novel outlier score criteria tailored to the structure of RPCA forest. The outlier score is the cornerstone of any outlier detection method, as it quantifies the degree to which a data point deviates from expected patterns. By designing new scoring mechanisms that leverage the unique properties of RPCA forest, one can potentially improve its performance. Additionally, one could investigate alternative methods for computing outlier scores, exploring how different formulations interact with the RPCA forest's architecture. These efforts could uncover new ways to enhance the model's sensitivity to outliers, particularly in challenging datasets with complex distributions. Furthermore, extending the RPCA forest to incorporate adaptive mechanisms, such as dynamic neighborhood sizes across different trees or weighted decision making for the splits instead of majority voting, could further boost its performance, making it a more effective tool for outlier detection and other related downstream tasks.

## Code and Data Availability

Full implementation of the proposed method in Python is available on Github:
https://anonymous.4open.science/r/Randomized-PCA-Forest-for-Outlier-Detection-76C4/
The datasets used for the experiments can be retrieved from:
http://www.dbs.ifi.lmu.de/research/outlier-evaluation/
For all the experiments, the first version of the datasets, without duplicates and with the least amount of outliers were used to ensure a realistic outlier detection scenario.

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
