# OpenReview forum: "Randomized PCA Forest for Unsupervised Outlier Detection"
_TMLR — Rejected by TMLR_

### Review · Reviewer_TMm8 · 2026-02-14

**Summary Of Contributions:**

The paper proposes RPCA Forest for unsupervised outlier detection, adapting Randomized PCA Forest (originally designed for approximate kNN search) into an anomaly detection method. The key idea is to construct RPCA trees using randomized PCA projections and define an outlier score that combines (i) a depth-based criterion and (ii) the mean distance of a point to other samples in its leaf node. The method is evaluated on 22 benchmark datasets and compared against classical approaches such as KNN, LOF, and Isolation Forest, as well as more recent methods like DAO.

**Audience:**

Yes

**Audience Explanation:**

Yes, the paper would likely interest a subset of TMLR readers, particularly those working on scalable anomaly detection, tree-based methods, or approximate nearest neighbor structures. The large empirical study and practical robustness are appealing.

However, the work is more empirical and engineering-oriented than theoretically novel. Readers looking for statistical guarantees, consistency results, or deeper theoretical insights may find the contribution limited.

**Broader Impact Concerns:**

The method is generally neutral but could be applied in sensitive domains such as fraud detection or medical diagnosis. The paper does not discuss risks such as misclassification of rare legitimate patterns or deployment without calibration. A short discussion of limitations and potential misuse would strengthen the submission.

**Claims And Evidence:**

Yes

**Claims Explanation:**

The empirical evaluation is extensive and generally well conducted. The method performs competitively across many datasets and often ranks among the top methods in terms of AUC. The use of statistical ranking tests (Friedman + post-hoc tests) is appropriate.

However, some claims are somewhat overstated. The paper repeatedly suggests superiority over state-of-the-art methods, yet performance varies across datasets and classical methods remain competitive in several cases. The evidence supports robustness and competitiveness, but not clear dominance.

Additionally, the scoring mechanism is heuristic. The multiplicative combination of depth-based and distance-based components is not theoretically justified, and no ablation study is provided to isolate the contribution of each term. This weakens the strength of the methodological claim.

**Requested Changes:**

The most important revision would be an ablation study separating the depth-only, distance-only, and combined scoring mechanisms. This is critical to justify the design choice.

The authors should also moderate claims of superiority and clarify the statistical significance and effect sizes more carefully.

Strengthening the theoretical discussion of the scoring function and providing a clearer derivation of the computational complexity would further improve the paper.

---

> ### Author Response · Authors · 2026-03-25
> **Author Response**
>
> Thank you for your insightful review. We agree that adding the ablation study
> will certainly help to clarify the decisions made for outlier score. In the new revision, Section 5.5
> includes an ablation study with complete experiments run on all datasets, as well as offering a
> statistical analysis and discussion to further shed light on the design choices.
>
> We also agree that the choice of words might have seemed a bit overstated, as in many cases, the
> proposed method shows competitive performance and not completely dominant figures. Hence,
> we reviewed the paper and revised the wording into a more moderate and informative version.
>
> As for the theoretical discussion and computational complexity, Section 3 in the new revision now
> includes a full overview of Randomized PCA. Section 4, which covers the proposed method, now
> includes a more detailed description of the Randomized PCA tree and how it is fit. We also improved
> Section 4.3 on computational complexity by adding more details.
>
> To address your suggestion and further shed light on the limitations and where the method might
> fail, we added theoretical discussions to Section 4, as well as a short paragraph, discussing the
> possible limitations, in Section 6.

---

### Review · Reviewer_t1MS · 2026-03-07

**Summary Of Contributions:**

This paper proposes an outlier detection method based on randomised PCA forest. They suggest a new method to compute outlier scores based on the depth of the leaf node and the distances to the other points in the same leaf node. The authors discuss computational and robustness properties of the method and conduct experiments showcasing the the performance compared to various baseline methods.

**Audience:**

Yes

**Audience Explanation:**

This paper surveys a large pool of outlier detection methods. I believe the experimental findings of the paper, if being theoretically grounded and clearly presented, would be of interest for the TMLR audience.

**Broader Impact Concerns:**

No impact concerns

**Claims And Evidence:**

No

**Claims Explanation:**

While the paper provides wide experimental evidence, the clarity of the exposition needs to be substantially improved.
The related work section lacks structure and sometimes gives too many details, which are nevertheless unclear.
There are descriptions of a wide range of methods, such as based on GANs, HDSVDD, HDIOD, and others, which do not appear in the later comparison. I believe condensing this section and adding some structure would greatly benefit the clarity.

Section 3, the proposed method, in contrast, lacks sufficient details. I would suggest to include separate notation section and to define all the notions used in the paper, such as randomized PCA and RPCA forest, formally.

The way Equations (2, 3) are introduced after Equation (1) lacks clear motivation. Could the method based on Equation (1) also be added to the experimental comparison?

The paper lacks any formal theoretical justification of the proposed method apart from the intuition that the outliers lie in the lower depths of the tree.

Finally, based on Figure 4 and 5, there are several examples where the proposed method is inferior to the prior work (e.g., on datasets D1, D2, D8, D22). These cases need to be separately discussed together with general failure scenarios of the proposed method.

**Requested Changes:**

The paper must be structured more clearly. The related work section can be condensed and only work most relevant to the current method explained in details. The method needs to be introduced more formally, with all used notions defined.

Some theoretical justification of the proposed method needs to be added, for example the motivation of Equation (2) from Equation (1).
Finally, the cases of failure of the proposed methodology need to be discussed.

---

> ### Author Response · Authors · 2026-03-25
> **Author Response**
>
> Thank you for your valuable comments and suggestions. We took your advice
> and added a detailed description of Randomized PCA in Section 3 in the new revision. In addition,
> Section 4 now includes a more detailed description of the Randomized PCA tree and how it is fit.
> We also improved Section 4.3 on computational complexity by adding more details.
>
> Following your valuable comment, we added an ablation study to the experiments to further justify
> our choices for the outlier score. In the new revision, Section 5.5 includes an ablation study with
> complete experiments run on all datasets, as well as offering a statistical analysis and discussion to
> further shed light on the design choices.
>
> We also agree that the related work section could use restructuring and shortening. Following your
> advice, we shortened and restructured this section to improve clarity and readability.
>
> Finally, thank you for your suggestion on the explanation required for the failure cases. In the
> new revision, we provide a discussion on the 4 cases in which RPCA forest fails, and insights into
> why this happens. In addition, we added theoretical discussions to Section 4, as well as a short
> paragraph discussing the possible limitations, in Section 6.

---

### Review · Reviewer_6dbF · 2026-03-19

**Summary Of Contributions:**

The paper proposes RPCA Forest, an unsupervised outlier detection method that adapts an existing approximate KNN search structure (RPCA Forest) for anomaly scoring using PCA components. The outlier score combines a depth-based criterion with a mean-leaf distance for better performance.

The authors then tests the performance of their algorithm on the 2016 benchmark by Campos and achieves good performance.

**Audience:**

Yes

**Audience Explanation:**

Outlier detection is a classical problem in machine learning and the motivations behind the need for further development do stand out. However, the paper in its current form is not publishable due to a lack of completeness in its experiments.

**Broader Impact Concerns:**

No immediate concern has been found.

**Claims And Evidence:**

No

**Claims Explanation:**

The algorithm present in the paper does not have any theoretical guarantee in any setting, and instead relies on experimental comparisons to establish the usefulness of their framework. However, compared to the state-of-the-art, the experiments are too limited to determine the extent of the usability of the proposed ideas.

One of the primary motivating arguments proposed in the paper is that existing outlier detection algorithms only fair only for "specific" kinds of outliers. However, the experiments do not look at the performance of the proposed framework for specific kinds, instead choosing to present aggregate results.

Furthermore, the paper does not use more recent benchmarks, instead focusing on the 2016 work by Campos et al. We refer the readers to a more recent benchmark [1] that evaluates a large set of outlier detection algorithm on specific kinds of outliers (such as local vs. global). In fact, the authors cite [1] but do not use their benchmark in the paper.

Finally, it should be noted that PCA has been recently used as an outlier detection algorithm in [2], where it is used to detect outliers specifically in data with ground truth clusters, with both theoretical and experimental support. It would be reasonable to compare the performance of RPCA forest with that of [2] in the datasets mentioned in [2] to further understand the global usefulness of RPCA forest (as it is one of the initial motivations mentioned in the paper).

[1] Songqiao Han, Xiyang Hu, Hailiang Huang, Minqi Jiang, and Yue Zhao. Adbench: Anomaly detection
benchmark. Advances in Neural Information Processing Systems (2022)

[2] Chandra Sekhar Mukherjee, Nikhil Deorkar, and Jiapeng Zhang. "Capturing the denoising effect of PCA via compression ratio." Advances in Neural Information Processing Systems 37 (2024)

**Requested Changes:**

The contributions of the paper will be clearer once experiments are done using more comprehensive and recent benchmarks that provide datasets containing specific class of outliers.

Specifically:

i) The authors should benchmark their algorithm using the Adbench [1] framework.

ii) The authors should compare their algorithm with [2], as it is a recent PCA-based outlier detection algorithm.

iii) The authors should consider proposing theoretical guarantees for any/some kinds of outliers.


[1] Songqiao Han, Xiyang Hu, Hailiang Huang, Minqi Jiang, and Yue Zhao. Adbench: Anomaly detection
benchmark. Advances in Neural Information Processing Systems (2022)

[2] Chandra Sekhar Mukherjee, Nikhil Deorkar, and Jiapeng Zhang. "Capturing the denoising effect of PCA via compression ratio." Advances in Neural Information Processing Systems 37 (2024)

---

> ### Author Response · Authors · 2026-03-25
> **Author Response**
>
> Thank you for your valuable comments. We based our evaluation on the com-
> prehensive survey by Campos et al., and followed their methodology to conduct experiments, with
> our additional extensive set of experiments and analyses to provide a robust and comprehensive
> empirical study of the proposed method and the competitors.
>
> ADBench also includes many of the methods that we already used in our experiments, however missing methods such as KNNW, DAO, and ODIN, should be integrated with ADBench. Many of the datasets used in ADBench are also similar to ours, selected from the study by Campos et al. (without giving information where the datasets are taken from and, hence, missing exact information on the version and preprocessing of the datasets), as well as a few different datasets. In order to run the experiments with ADBench, we would need a large scale repetition of many parts of our already-conducted empirical evaluation, which would take an unnecessarily long amount of time. As we believe our current experiments are comprehensive and extensive, and we are covering many
> relevant methods and datasets which are also included in the ADBench, we do not find repeating the experiments with ADBench fruitful for this paper. We however thank you for your suggestion.
>
> As for the suggested reference [2], we find the paper very interesting as it also uses PCA for outlier detection, but with a completely different and novel approach. We integrated the implementation of the paper with our benchmark and tried running the algorithm on our datasets. However, we observed extensive memory usage patterns and a very low scalability, especially with regards to n. As
> there was no time complexity reported in the paper, we used the provided python implementation
> to get a better understanding of its complexity. The time complexity is dominated by the distance matrix calculation, which is performed both in the original feature space and the reduced PCA space:
>
> • In the original space with d dimensions, computing the distance between all pairs requires O(n^2 · d) operations.
>
> • While the PCA projection itself takes O(nd · min(n, d)), the subsequent distance calculation in the k-dimensional PCA space adds O(n^2 · k).
>
> • Finally, calculating the variance for each point involves iterating through each row of the n×n matrix, adding O(n^2).
>
> Consequently, the total time complexity is approximately O(n^2 · d). This complexity makes the
> method very interesting for high-dimensional datasets, like single-cell RNA-seq data, but it is very
> heavy for big data and large values of n. As observed in [2], the largest n this paper covers in their
> experiments is 6,498. In our experiments we have two datasets with around 50,000 samples. Due
> to the very high time-complexity of [2] with regards to n, it is not feasible to integrate it fully with
> our experiments. However, as the method was theoretically very interesting, we decided to include
> it in our related work section.
>
> For the theoretical guarantees, it is difficult with respect to the nature of the method, to provide
> theoretical guarantees. However, we agree that the paper can benefit from more theoretical discussions, as pointed out by all reviewers. Hence, we added some theoretical discussions on our expectations to Section 4.1, and the limitations in Section 6.

---

### Author Response · Authors · 2026-03-25
**Revision**

A new version of the paper addressing the valuable comments of the reviewers is now available.

---

### Decision · Action_Editor_Qc9T · 2026-05-02

**Recommendation:** Reject

**Audience:**

Yes

**Audience Explanation:**

Outlier detection is a classical and highly relevant problem in machine learning. The paper’s focus on scalable, tree-based anomaly detection and approximate nearest neighbor structures would certainly appeal to TMLR readers interested in practical, engineering-oriented solutions.

However, as noted in the previous assessment, this potential interest is heavily bottlenecked by the current draft's execution. The findings will only be of tangible value to the community once the critical gaps in empirical evaluation (e.g., missing modern benchmarks and comparison with more recent approaches), the lack of theoretical justification, and the structural presentation issues are fully addressed.

**Claims And Evidence:**

No

**Claims Explanation:**

This paper introduces an unsupervised outlier detection method that adapts the Randomized Principal Component Analysis (RPCA) Forest (a technique originally designed for approximate k-NN search). The proposed outlier score is computed using a combination of leaf-node depth and mean-leaf distance.

During the review process, all three reviewers raised significant concerns. First, the method lacks theoretical justification, e.g., the scoring mechanism relies heavily on heuristics. Second, the submission makes overstated claims regarding its empirical performance. Specifically, it fails to demonstrate state-of-the-art results on several datasets and critically omits evaluations on the latest benchmarks (such as ADBench [1]) and comparisons against recent baselines (e.g., [2]). Finally, the manuscript requires substantial structural and stylistic revisions; the related work section is disproportionately lengthy, while the methodology section lacks necessary technical depth.

While the authors provided a rebuttal, it only partially addressed the reviewers' critiques. The revised manuscript still lacks rigorous theoretical grounding and strong empirical evidence against modern approaches. More critically, the revision expanded the page count by 50% compared to the original submission, introducing new presentation issues. Structural inconsistencies emerged (e.g., Algorithm 0 is now presented as in-line text rather than a formatted block), and reviewers suggested that much of the new content should be moved to the appendix to improve readability.

Overall, the Action Editor agrees with the reviewers' concerns. The current manuscript lacks the theoretical rigor and modern empirical evaluation expected for publication. Furthermore, the 50% increase in length and the associated formatting inconsistencies indicate that the paper requires another round of major revision and careful peer review.

[1] ADBench: Anomaly Detection Benchmark. NeurIPS 2022.

[2] Capturing the Denoising Effect of PCA via Compression Ratio. NeurIPS 2024.